# The antioxidant betulinic acid enhances porcine oocyte maturation through Nrf2/Keap1 signaling pathway modulation

Min Ju Kim[1,2☯], Hyo-Gu Kang[1,3☯], Se-Been Jeon[1,2], Ji Hyeon Yun[1,4], Pil-Soo Jeong[1], Bo-Woong Sim[1,5], Sun-Uk Kim[1,5], Seong-Keun Cho[2,6]*, Bong-Seok Song[1]*

1 Korea Research Institute of Bioscience and Biotechnology, Futuristic Animal Resource & Research Center, Cheongju, Republic of Korea, 2 Department of Animal Science, College of Natural Resources & Life Science, Pusan National University, Miryang, Republic of Korea, 3 Department of Animal Science and Biotechnology, College of Agriculture and Life Science, Chungnam National University, Daejeon, Republic of Korea, 4 Department of Animal BioScience, School of Animal Life Convergence, Hankyong National University, Ansung, Republic of Korea, 5 Department of Functional Genomics, University of Science and Technology, Daejeon, Republic of Korea, 6 Department of Animal Science, Life and Industry Convergence Research Institute, College of Natural Resources and Life Science, Pusan National University, Miryang, Republic of Korea

☯ These authors contributed equally to this work.
* skcho@pusan.ac.kr (S-KC); sbs6401@kribb.re.kr (B-SS)

**Data Availability Statement:** All relevant data are within the manuscript and its Supporting Information files.

## Abstract

During *in vitro* maturation, excess levels of reactive oxygen species (ROS) are a major cause of developmental defects in embryos. Betulinic acid (BA) is a naturally produced antioxidant in white birch bark. Recent studies have shown that BA exhibits antioxidant properties in various cells through the activation of antioxidant genes. Therefore, we investigated the effect of BA treatment on porcine oocytes and its underlying mechanism during oocyte maturation. Treatment with 0.1 µM BA significantly increased the proportion of MII oocytes compared with controls, and BA-treated oocytes had significantly higher development rates, trophectoderm cell numbers, and cell survival rates than controls. These results demonstrate that BA treatment improved the developmental competence of oocytes. Following BA treatment, oocytes exhibited reduced ROS levels and elevated glutathione (GSH) levels, accompanied by the enhanced expression of antioxidant genes, compared with control oocytes. To evaluate the antioxidant effects of BA, oocytes were exposed to $H_2O_2$, a potent ROS activator. Impaired nuclear maturation, ROS levels, and GSH levels induced in oocytes by $H_2O_2$ exposure was restored by BA treatment. As these antioxidant genes are regulated by the Nrf2/Keap1 signaling pathway, which is involved in antioxidant responses, we applied the Nrf2 inhibitor brusatol to investigate the effects of BA on this pathway. The negative effects of brusatol on meiotic maturation and oocyte quality, including levels of ROS, GSH, and antioxidant-related gene expression, were mitigated by BA treatment. Our results suggested that BA plays an effective role as an antioxidant in porcine oocyte maturation through adjusting the Nrf2/Keap1 signaling pathway. This finding provides valuable insights into the mechanisms governing oocyte maturation and embryonic development.

**Funding:** The author(s) received no specific funding for this work.

## Introduction

Assisted reproductive technology (ART) has shown significant progress during the past decade. However, improper culture conditions can result in the production of poor-quality oocytes, leading to failures in ART outcomes [1]. Therefore, an appropriate *in vitro* culture system must be established to support proper oocyte maturation, which would ultimately improve the success rates of ART procedures [2]. However, optimizing *in vitro* maturation (IVM) conditions for oocyte culture remains challenging, with oxidative stress emerging as a major cause of oocyte maturation defects [3,4].

Oxidative stress is induced by reactive oxygen species (ROS), which are typically generated by mitochondria as cellular metabolites and play essential roles in modulating cellular proliferation [5]. Although ROS are essential for the maintenance of certain normal physiological functions, excessive ROS levels can induce oxidative stress, reducing cellular proliferation [6]. Excess ROS levels during oocyte maturation can also cause embryo arrest, DNA damage, and alterations in gene expression, leading to embryonic defects [7]. Therefore, regulation of redox signaling to reduce oxidative stress is an important goal in IVM systems optimization [8]. Despite efforts to develop methods for oxidative stress alleviation, its levels remain high under typical IVM conditions [9]. *In vivo* embryos are protected from oxidative stress by oxygen scavengers within follicular and oviduct fluids, whereas *in vitro* oocytes rely on antioxidant defense mechanisms to mitigate oxidative damage [10]. Thus, the enhancement of oocyte maturation during IVM may involve utilizing various chemicals to reduce ROS levels [11].

Betulinic acid (BA, 3b-hydroxy-lup-20(29)-en-28-oicacid) is a natural antioxidant found in white birch bark and rosemary, among other plant sources [12]. BA has a broad spectrum of pharmacological and biochemical effects conferring anti-cancer, anti-malarial, anti-bacterial, anti-inflammatory, and immunomodulatory activity [12]. The mechanism underlying BA biochemical activity involves the inhibition of topoisomerase, which causes cellular apoptosis [13]. However, at lower dosages, BA demonstrates antioxidant activity across various cell types by upregulating antioxidant genes such as *superoxide dismutase* (*SOD*), *catalase* (*CAT*), *glutathione peroxidase1* (*GPX1*), and *heme oxygenase-1* (*HO-1*) [14–16]. Recent studies have revealed that BA can enhance the protein expression of nuclear factor erythroid-2-related factor 2 (Nrf2) in H9c2 cells, suggesting that its antioxidative activity is mediated through the modulation of Nrf2 pathways [17].

The Nrf2/Keap1 signaling pathway acts as central defense mechanism against oxidative stress, regulating the expression of antioxidant enzymes and genes involved in the redox system [18]. Under normal physiological conditions, Nrf2 undergoes ubiquitination by Keap1 in the cytoplasm, leading to subsequent degradation via the proteasome [19]. However, under excess oxidative stress, Nrf2 separates from Keap1 to migrate into the nucleus [19], where it binds to antioxidant response elements, to modulate the expression of antioxidant elements such as *SOD*, *CAT*, *GPX1*, and *HO-1*, thereby regulating the antioxidant defense mechanisms of various cell types [20].

Despite the recognized association between BA and antioxidant responses mediated by the Nrf2/Keap1 signaling pathway, the effects of BA during IVM remain unclear. Therefore, we investigated this signaling pathway using porcine oocytes, which are important biomedical models due to their physiological similarity to human oocytes [21]. The objective of this study was to clarify the mechanisms by which BA alleviates oxidative stress via activation of the Nrf2/Keap1 signaling pathway during oocyte maturation.

## Materials and methods

### Chemicals

Unless specifically clarified, all chemicals and reagents were procured from Sigma Aldrich Korea (St. Louis, MO, USA).

## Oocyte collection

Ovaries sourced from porcine were procured from a local slaughterhouse and transported with under controlled conditions proper temperature range of in 0.9% saline solution to the laboratory. For prevent contamination, saline solution supplemented with 0.75 μg/mL benzyl-penicillin potassium (Wako, Osaka, Japan) and 0.5 μg/mL streptomycin sulfate salt. After cleansing ovaries, extracted cumulus-oocyte complexes (COCs) carefully isolated from each 3 to 8 mm in diameter follicles, using syringe. COCs which has the sufficient layers of cumulus cells with even cytoplasm were chosen for IVM. COCs washed with 0.9% saline solution supplemented with 1 mg/mL bovine serum albumin (BSA).

## Chemical treatment

BA and brusatol (Bru) dissolved using dimethyl sulfoxide (DMSO), and $H_2O_2$ dissolved using $H_2O$. All dissolved chemical diluted to a final concentrations of 0.01, 0.1, 1 μM (BA), 1 mM ($H_2O_2$), and 30 nM (Bru) with IVM medium. Concentration of $H_2O_2$ and Bru correspond to previous studies [18,22]. All chemical treated full time of IVM (44 h). In addition, all solvent was added less than 0.1% in IVM medium. BA purchased from Sigma Aldrich Korea; 855057.

## In vitro maturation (IVM)

The cleansed COCs were cultured with IVM medium contain or absence BA for a 44 hours at 38.5°C with an atmosphere of 5% $CO_2$ in air. Throughout the initial maturation phase (0–22 h, IVM I), the IVM I medium was formulated with 10% porcine follicular fluid, 0.57 mM cysteine, 25 μM β-mercaptoethanol, 10 ng/mL epidermal growth factor, 10 IU/mL pregnant mare serum gonadotropin, and 10 IU/mL human chorionic gonadotropin. Subsequently, a second maturation phase (from 22 to 44 h, IVM II) was initiated, utilizing the same medium formulation although without hormonal supplementation. After IVM, matured COCs are denuded by 0.1% hyaluronidase. Metaphase II oocytes were selected by presence of polar body.

## Parthenogenetic activation and in vitro culture (IVC)

Matured oocytes that denuded by hyaluronidase are washed for parthenogenetic activation. For activation, 15 μM of ionomycin was diluted with Dulbecco's phosphate-buffered saline (DPBS; Gibco, Carlsbad, CA, USA) for 5 min in dark. For prevent contamination, DPBS added with 60 μg/mL gentamicin sulfate salt, 75 μg/mL streptomycin sulfate, and 4 mg/mL BSA. After the treatment ionomycin, oocytes were washed and cultured in porcine zygote medium-3 with 4 mg/mL BSA (IVC medium) containing with 5 μg/mL cytochalasin B and 2 mmol/L 6-dimethylaminopurine at 38.5°C with an atmosphere of 5% $CO_2$ in air for 4 h. Subsequently, activated oocytes were transferred to fresh IVC medium and incubated at 38.5°C with an atmosphere of 5% $CO_2$ in air for 6 d.

## Terminal deoxynucleotidyl transferase dUTP-digoxygenin nick end labeling (TUNEL) assay

After 6 d, cultured blastocysts are used for detecting apoptosis rate. Therefore, for TUNEL assay, an In Situ Cell Death Detection Kit (Roche, Basel, Switzerland) was used. Collected blastocysts soaked with 4% (v/v) paraformaldehyde for fixation after 6 d of culture. After fixation, blastocysts were washed three times with DPBS supplemented with 0.1% PVA (PBS-PVA). For permeabilization, 1% (v/v) Triton X-100 was used, and washed three times after. Washed blastocysts were incubated with or without TUNEL enzyme for 1 h at 38.5°C. Not stained sample was used for negative control. Stained blastocysts were washed with PBS-PVA and mounted

on slide glasses (Marienfeld, Lauda-Königshofen, Germany) with DAPI (Vector Laboratories Inc., Burlingame, CA, USA), the mounting solution. Mounted blastocysts were observed under a fluorescence microscope (Leica), and the numbers of apoptotic cells were counted.

## Immunofluorescence

Fixed blastocysts and oocytes with formalin solution washed with PVA-PBS three times. 1% (v/v) Triton X-100 was used for permeabilization and subsequently washed three times in. At RT, washed samples were moved to a blocking solution for 1 h. After, each samples were incubated for 4˚C overnight with proper primary antibodies following CDX2 (AM392; BioGenex, Fremont, CA, United States), Nrf2 (1:200; ab31163; Abcam, Cambridge, MA, United States) and Keap1 (1:200; ab226997; Abcam). The next day, samples were washed three times and move into blocking solution same as first blocking step. Afterwards, with proper diluted secondary antibody, sample were incubate 1 h at RT (1:200; A11008, A11012, A11029; Invitrogen). Finally each stained samples were washed and mounted with DAPI on slide glass and observed with a fluorescence microscope, quantified with ImageJ software (version 1.47; National Institutes of Health, Bethesda, MD, USA). All results are normalized to the control beside CDX2.

## Measurement of intracellular ROS and GSH levels

Measurement of intracellular ROS and GSH in embryos was performed as described previously [23]. Briefly, CM-H2DCFDA (Invitrogen, Carlsbad, CA, USA) and CMF2HC (Invitrogen) was used for detection of ROS and GSH as green and blue fluorescence. Oocytes from each group were incubated for 10 min and 30 min in a solution of PBS-PVA mixed with 5 μM CM-H2DCFDA and 10 μM CMF2HC. Incubated oocytes were washed with PBS-PVA and move into 20 μL droplets of PBS-PVA. Stained oocytes were observed under a fluorescence microscope with ultraviolet filters (460 nm; ROS, 370 nm; GSH). Individual oocytes of fluorescence intensities were quantified with ImageJ software. All results are normalized to the control.

## Quantitative real-time polymerase chain reaction (qRT-PCR)

Extraction of poly(A) mRNA and synthesis of cDNA was described previously [23]. Briefly, using Dynabeads mRNA Direct Micro Kit (Invitrogen), mRNA from oocyte were extracted from in each group. Oocytes was lysis with lysis buffer and separate the mRNA using Dynabeads oligo $(dT)_{25}$. Dynal magnetic bar (Invitrogen) was used for separated from the binding buffer. Buffers A and B was used for washing the beads with poly(A) mRNAs. Tris buffer was used for separation bead and poly(A) mRNA and the resulting poly(A) mRNAs were reverse-transcribed. For genomic DNA was elimination and cDNA synthesis, Prime Script RT Reagent Kit was used (Takara Bio Inc., Shiga, Japan). After genomic DNA erase stage at RT, samples were incubated for cDNA synthesis. Using the Mx3000P qPCR system, qRT-PCR was performed (Agilent Technologies, Santa Clara, CA, USA) with SYBR premix Ex Taq (Takara Bio Inc.). Gene expression was expressed in terms of the fold change, and the 2Δ(SDCTΔCDCT) method was used to analyze gene expression. The primers used in the study are listed in Additional file 1: S1 Table.

## Statistical analyses

Each experiment was conducted a minimum of three times, and the results are expressed as means ± standard error of the mean. Factorial ANOVA was employed followed by Duncan's

multiple range test conducted using SigmaStat (Systat Software Inc., San Jose, CA, USA). Statistical significance was determined for p-values less than 0.05.

### Ethics statement

All procedures and use of pigs were approved by the Korea Research Institute of Bioscience and Biotechnology (KRIBB) Institutional Animal Care and Use Committee (Approcal No. KRIBB-AEC-24171)

## Results

### Enhancement of nuclear maturation and developmental competence in BA-treated porcine oocytes

To investigate the effects of BA during oocyte maturation, porcine oocytes were cultured with various concentration of BA (0.0, 0.01, 0.1, and 1.0 μM) during IVM. BA treatment significantly enhanced oocyte maturation rates and decreased the numbers of immature oocytes following IVM compared to the control (Fig 1A and 1B and S2 Table). Oocytes treated with BA and cultured after parthenogenetic activation for 6 days showed significant increases in blastocyst formation rates and total cell numbers in the 0.1 μM BA treatment group (Fig 1C–1F and S3 Table). Notably, the blastocyst formation rate was significantly lower under treatment with 1.0 μM BA compared to 0.1 μM BA. Therefore, 0.1 μM BA was selected for subsequent experiments. A TUNEL assay revealed that treatment with 0.1 μM BA significantly reduced the apoptosis rate, showing fewer apoptotic cells (Fig 1G–1I and S4 Table), and increased TE cell numbers although no significant differences were observed in ICM cell numbers (Fig 1J–1L and S5 Table). These results suggest that BA treatment during IVM enhances meiotic progression, thereby promoting successful embryonic development.

### Effects of BA on porcine oocytes under oxidative stress

To explore whether the enhanced oocyte maturation rates and embryonic developmental competence observed following BA treatment were related to its antioxidant properties, we assessed intracellular ROS levels in matured oocytes. BA-treated oocytes had significantly lower ROS levels compared to the control (Fig 2A and 2B). Additionally, intracellular GSH levels were significantly higher following BA treatment (Fig 2C and 2D). BA treatment also significantly increased the expression levels of the antioxidant-related genes *SOD1*, *SOD2*, *CAT*, *GPX1*, and *HO-1*. These results indicate that BA can reduce intracellular oxidative stress levels by regulating antioxidant genes.

### Effects of BA on oocyte maturation and developmental competence under $H_2O_2$ treatment

We evaluated the antioxidant efficacy of BA using $H_2O_2$, which is a potent ROS activator. Treatment with $H_2O_2$ led to a significant decrease in the proportion of maturate oocytes (Fig 3A and 3B and S6 Table). However, co-treatment of $H_2O_2$-treated oocytes with 0.1 μM BA resulted in a significant increase in oocyte maturation rates compared to oocytes treated with $H_2O_2$ alone (Fig 3A and 3B and S6 Table). Furthermore, the decreased cleavage and blastocyst formation rates but not total cell numbers induced by $H_2O_2$ treatment were significantly restored oocytes upon BA treatment (Fig 3C–3F and S7 Table). A TUNEL assay revealed that the apoptosis rate induced by $H_2O_2$ treatment was significantly ameliorated by BA treatment (Fig 3G–3I and S8 Table). Additionally, the altered numbers of TE cells and ICM cells compared to the control became similar under BA treatment (Fig 3J–3L and S9 Table). These

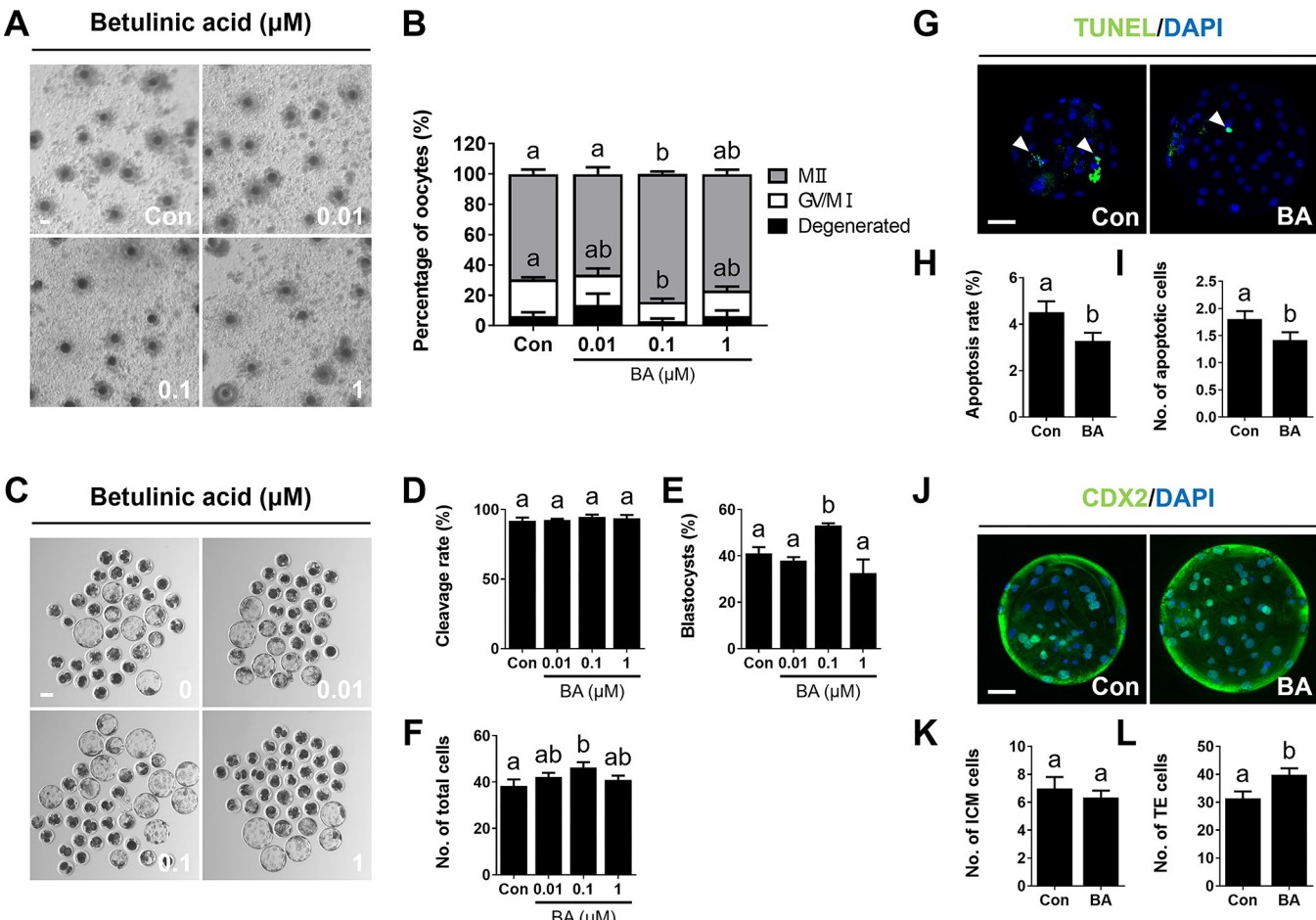

**Fig 1. Betulinic acid (BA) treatment enhances porcine oocytes maturation. A** Bright-field images of oocytes after 44 h of *in vitro* maturation (IVM) with various concentrations of BA. Bar = 100 μm. **B** Proportions of different stages of nuclear maturation (0.0 μM BA, n = 264; 0.01 μM BA, n = 243; 0.1 μM BA, n = 264; 1.0 μM BA, n = 274). **C** Bright-field images of BA-treated oocytes at 6 days of culture after parthenogenetic activation. Bar = 100 μm. **D–F** Cleavage rate, blastocyst formation rate, and total cell number measurements (0.0 μM BA, n = 192; 0.01 μM BA, n = 124; 0.1 μM BA, n = 210; 1 μM BA, n = 216). **G** Representative images of blastocysts from a terminal deoxynucleotidyl transferase dUTP nick-end labeling (TUNEL) assay. Merged images show cells positive for 4′,6-diamidino-2-phenylindole (DAPI; blue) and TUNEL (green; white arrows). Bar = 50 μm. **H, I** Apoptosis rates and cell numbers (n = 46 per group). **J** Representative images of CDX2 labeling in blastocysts. Merged images show cells positive for DAPI (blue) and CDX2 (green). Bar = 50 μm. **K, L** Numbers of trophectoderm (TE) and inner cell mass (ICM) cells (n = 33 per group). Data are means of three independent experiments; different letters indicate significant differences (P < 0.05).

results suggest that BA treatment can ameliorate abnormal developmental competence resulting from impaired oocyte maturation, potentially by reducing ROS levels induced by $H_2O_2$ via its antioxidant properties.

## Effects of BA on $H_2O_2$-exposed porcine oocytes under oxidative stress

To investigate the antioxidant effects of BA on $H_2O_2$-treated oocytes, we measured intracellular ROS and GSH levels. Elevated ROS levels induced by $H_2O_2$, compared to the control, were significantly reduced by BA treatment (Fig 4A and 4B). Additionally, the reduction in GSH levels induced by $H_2O_2$, compared to the control, were significantly restored by BA treatment (Fig 4C and 4D). As previously demonstrated, BA treatment significantly upregulated the expression levels of antioxidant genes in porcine oocytes (Fig 2E), and the decrease in antioxidant gene expression induced by $H_2O_2$ was significantly reversed by BA compared to the

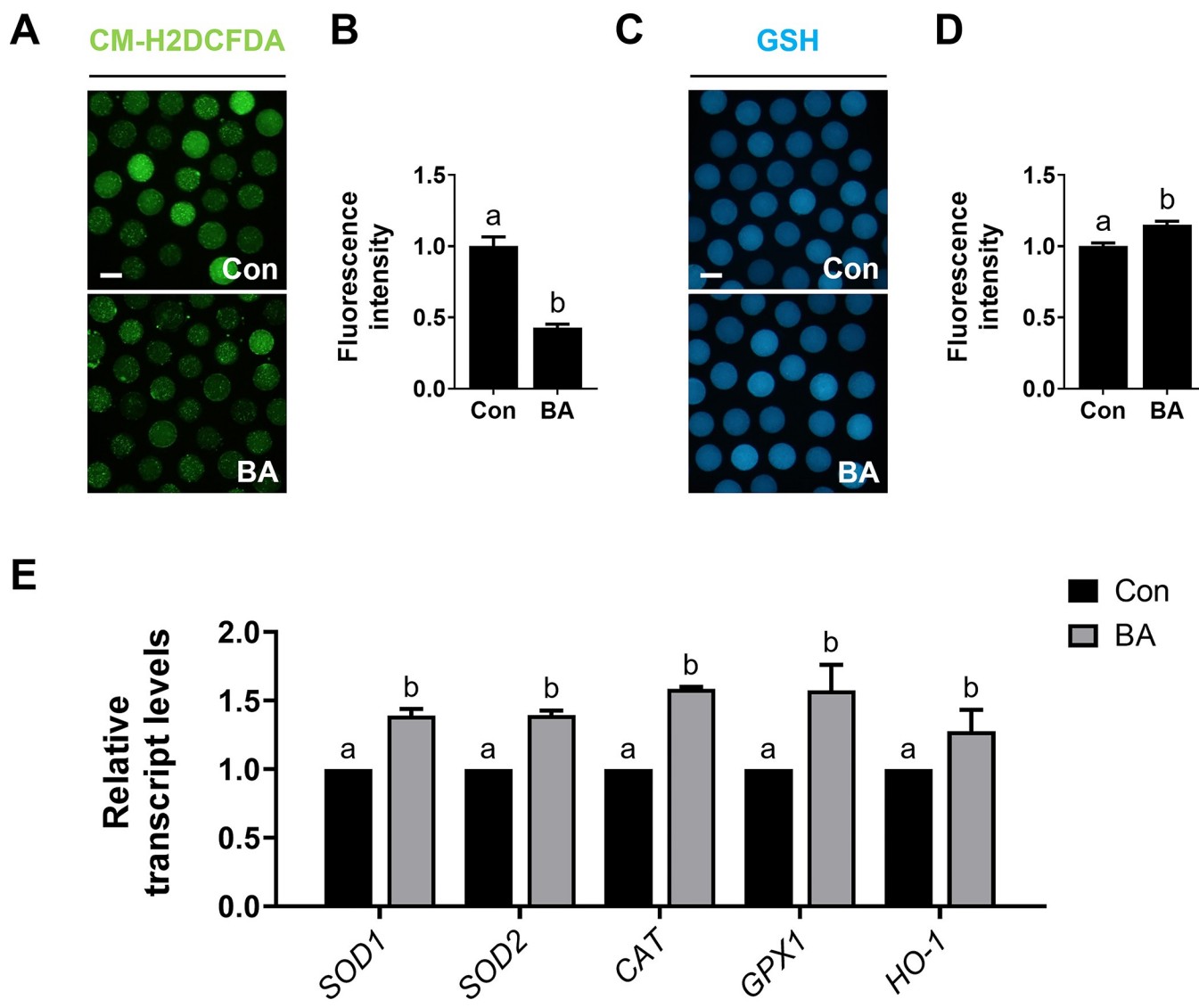

**Fig 2. BA treatment alleviates oxidative stress in porcine oocytes. A** Representative images of CM-H2DCFDA staining (green) in oocytes after 44 h of IVM. Bar = 100 μm. **B** Measurement of reactive oxygen species (ROS) fluorescence intensity (n = 40 per group). **C** Representative images of glutathione (GSH; blue) staining in oocytes after 44 h of IVM. Bar = 100 μm. **D** Measurement of GSH fluorescence intensity (n = 40 per group). **E** Transcription levels derived from quantitative reverse-transcription polymerase chain reaction (qRT-PCR) analysis of antioxidant genes in oocytes (n = 3 per group). Data are means of three independent experiments; different letters indicate significant differences ($P < 0.05$).

control (Fig 4E). As these genes are regulated by the Nrf2/Keap1 signaling pathway, we also examined the gene expression of *Nrf2* and *Keap1*. The results revealed that the decreased expression of *Nrf2* and increased expression of *Keap1* induced by $H_2O_2$ were significantly mitigated by BA treatment (Fig 4E). These results show that BA can ameliorate elevated intracellular oxidative stress levels in $H_2O_2$-treated oocytes through the regulation of antioxidant genes, under the control of the Nrf2/Keap1 signaling pathway.

### Effects of BA on the Nrf2/Keap1 pathway in Bru-exposed porcine oocytes

Given the evidence suggesting the potential involvement of BA in the Nrf2/Keap1 signaling pathway, we examined the levels of Nrf2 and Keap1 proteins in oocytes following IVM. We

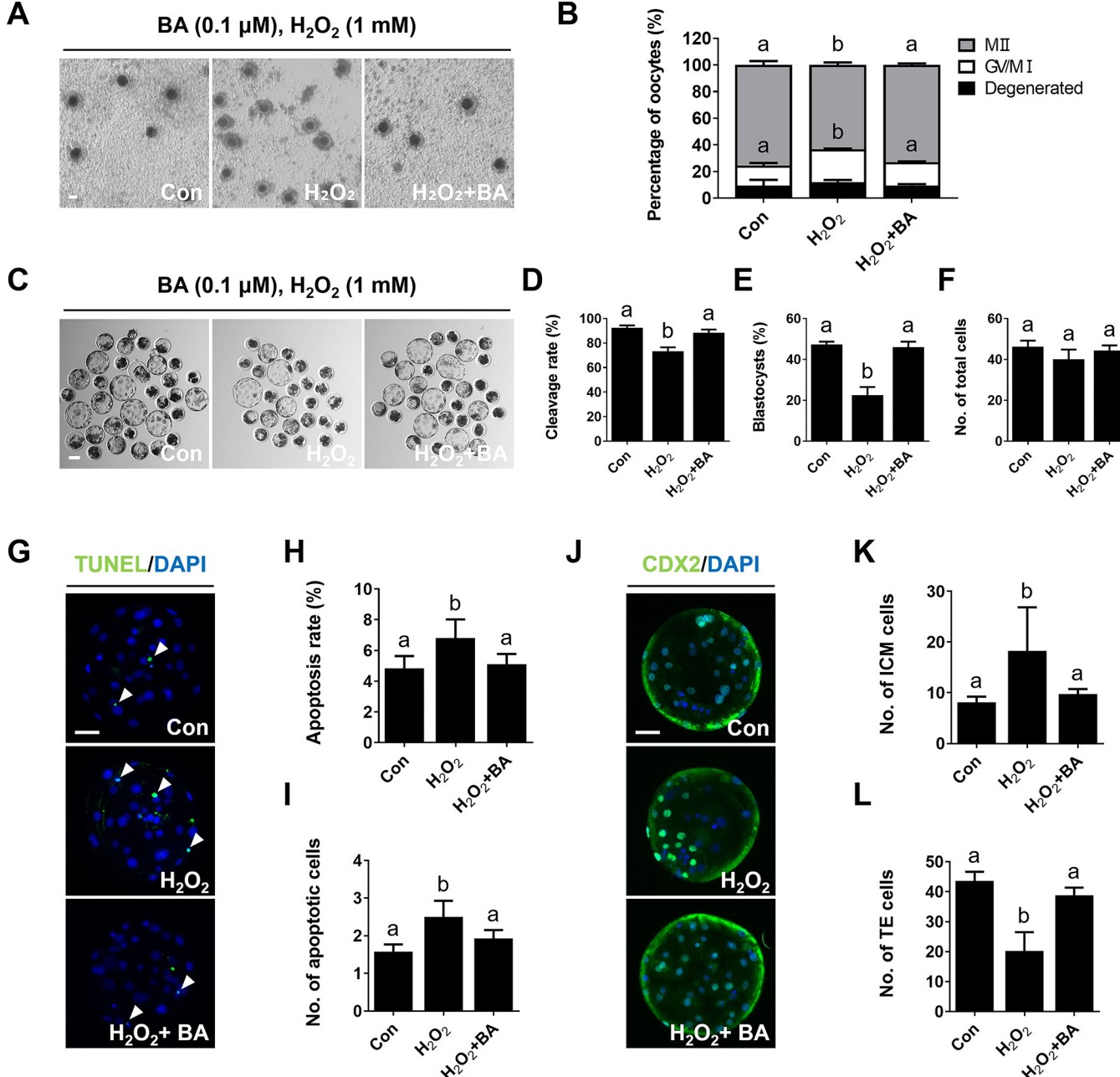

**Fig 3. BA treatment restores impaired oocytes maturation caused by $H_2O_2$ exposure in porcine oocytes. A** Bright-field images of oocytes after IVM. Bar = 100 μm. **B** Proportions of different stages of nuclear maturation (0.0 μM BA, n = 151; $H_2O_2$, n = 151; $H_2O_2$+BA, n = 149). **C** Bright-field images of BA-treated oocytes at 6 days of culture after parthenogenetic activation. Bar = 100 μm. **D–F** Cleavage rate, blastocyst formation rate, and total cell number measurements (0.0 μM BA, n = 116; $H_2O_2$, n = 86; $H_2O_2$+BA, n = 105). **G** Representative images of TUNEL labeling in blastocysts. Merged images show cells positive for DAPI (blue) and TUNEL (green; white arrows). Bar = 50 μm. **H, I** Apoptosis rates and cell numbers (n = 23 per group). **J** Representative images of CDX2 labeling in blastocysts. Merged images show cells positive for DAPI (blue) and CDX2 (green). Bar = 50 μm. **K, L** Numbers of TE and ICM cells (n = 23 per group). Data are means of three independent experiments; different letters indicate significant differences (P < 0.05).

used Bru, a potent Nrf2 inhibitor, to inhibit the Nrf2/Keap1 signaling pathway, and assessed whether BA could rescue this inhibition through its activation. The results revealed that BA treatment significantly increased the levels of nucleus Nrf2 compared to the control, whereas Bru treatment reduced nuclear Nrf2 levels (Fig 5A and 5B). Furthermore, the decreased levels

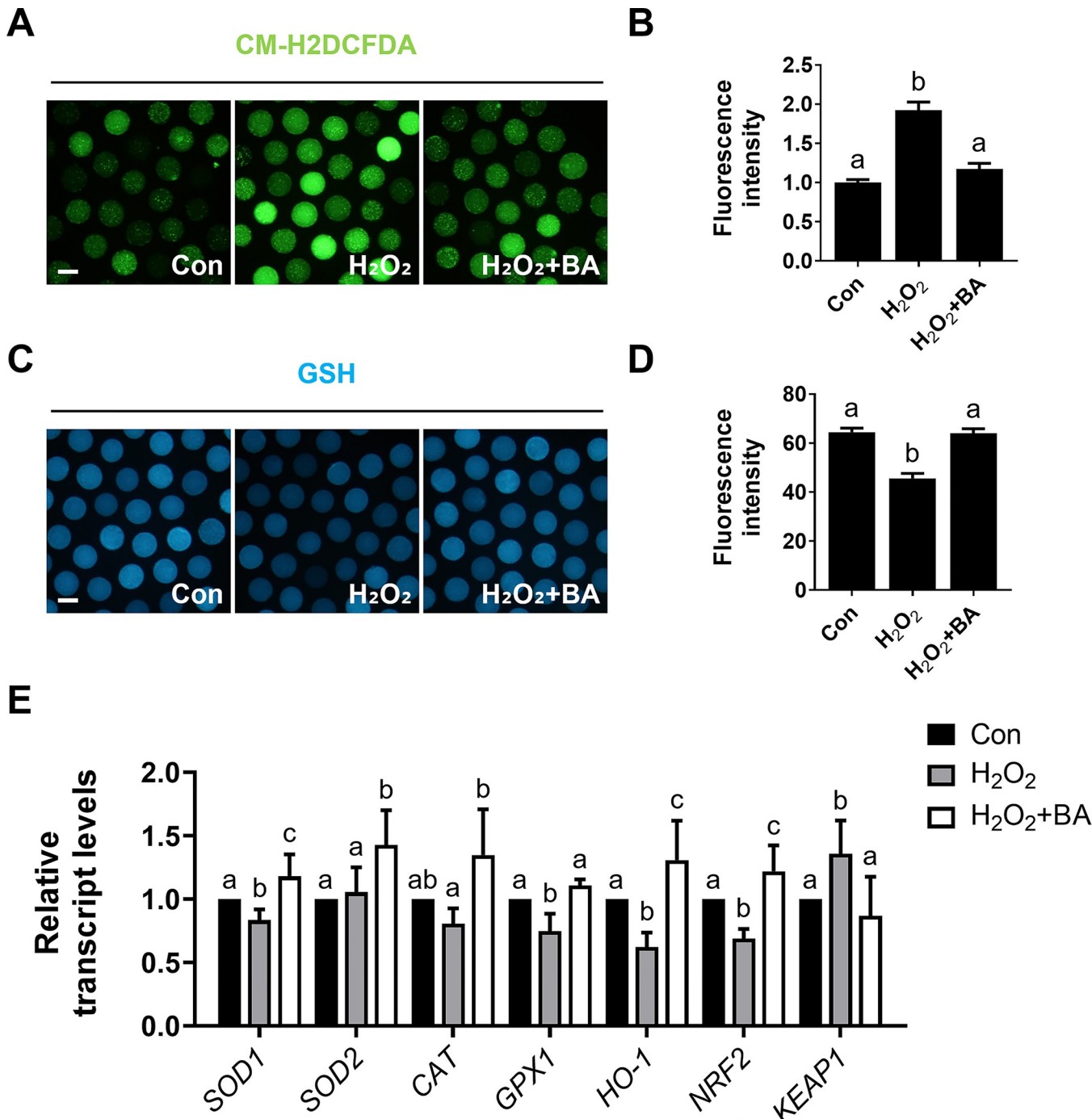

**Fig 4. BA treatment restores impaired oxidative stress caused by $H_2O_2$ in porcine oocytes.** **A** Representative images of CM-H2DCFDA staining (green) in oocytes after 44 h of IVM. Bar = 100 μm. **B** Measurements of ROS fluorescence intensity (n = 40 per group). **C** Representative images of GSH staining (blue) in oocytes after 44 h of IVM. Bar = 100 μm. **D** Measurements of GSH fluorescence intensity (n = 40 per group). **E** Transcription levels derived from qRT-PCR analysis of antioxidant genes and Nrf2/Keap1 signaling pathway-related genes in oocytes (n = 3 per group). Data are means of three independent experiments; different letters indicate significant differences (P < 0.05).

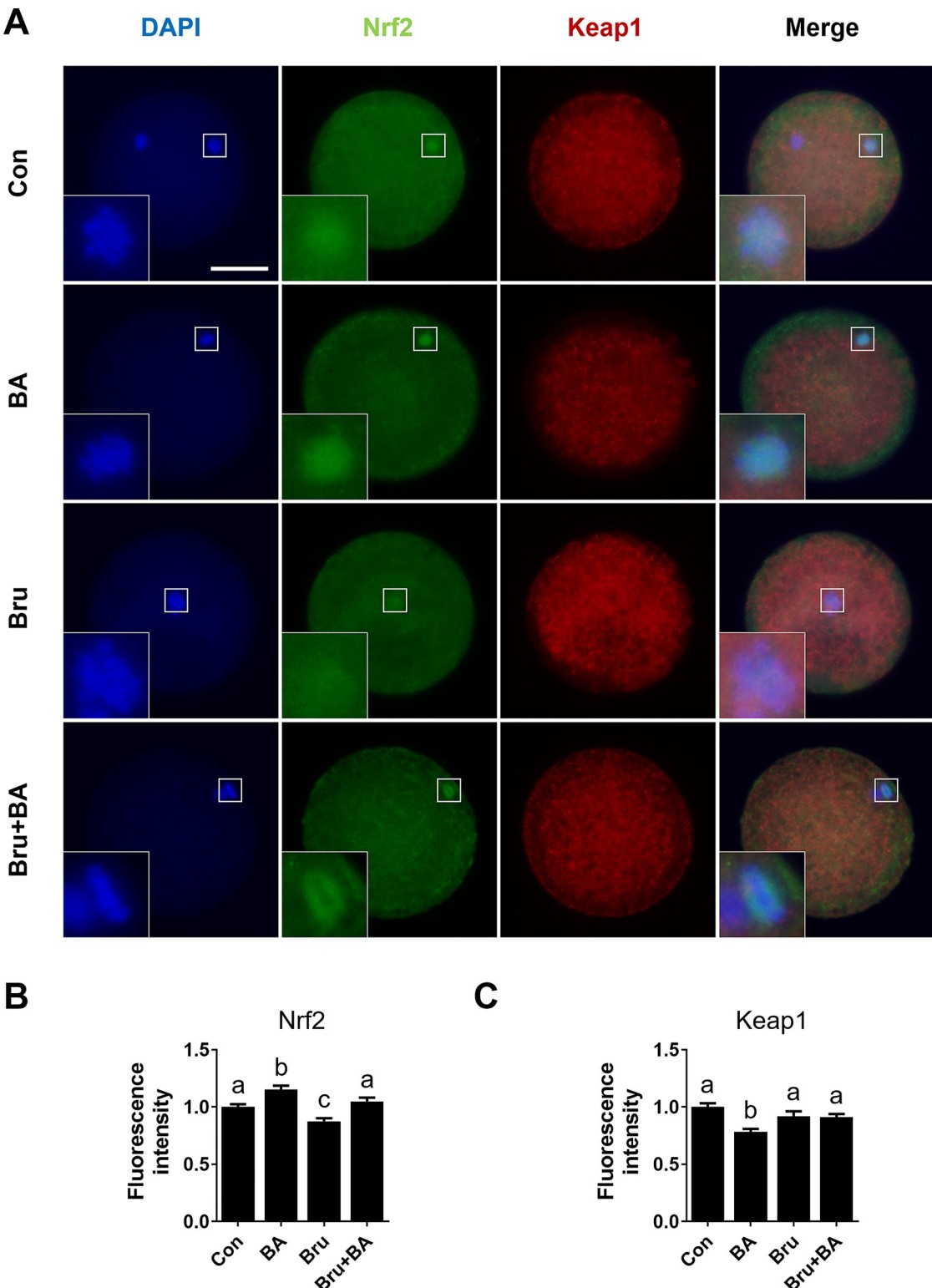

**Fig 5. BA treatment restores abnormal expression of Nrf2/Keap1 proteins caused by brusatol (Bru). A** Representative images of oocytes stained for Nrf2 and Keap1, and merged images (n = 27 per group). Merged images show cells positive for DAPI (blue), Nrf2 (green), and Keap1 (red). White box indicates the magnified nuclear region. **B** Nrf2 and **C** fluorescence intensity measurements. Data are means of three independent experiments; different letters indicate significant differences (P < 0.05).

of nuclear Nrf2 induced by Bru were significantly restored by BA treatment (Fig 5A and 5B). However, cytoplasmic levels of Keap1 were significantly reduced by BA treatment, whereas Bru treatment showed no significant difference (Fig 5A and 5C). These results suggest that BA treatment promotes the dissociation of Nrf2 from Keap1, facilitating translocation to the nucleus and promoting antioxidant gene expression.

### Effects of BA on porcine oocyte maturation and developmental competence under Bru exposure

Treatment with Bru significantly reduced the proportion of maturated oocytes (Fig 6A and 6B and S10 Table). However, co-treatment of Bru-treated oocytes with 0.1 μM BA resulted in a significant increase in the proportion of mature oocytes compared to Bru-treated oocytes (Fig 6A and 6B and S10 Table). Additionally, the decreased blastocyst formation rate and total cell numbers of oocytes following Bru treatment were significantly rescued by BA treatment (Fig 6C–6F and S11 Table). The increased apoptosis rate induced by Bru treatment was significantly ameliorated by BA treatment (Fig 6G–6I and S12 Table). Moreover, the altered numbers of TE cells became comparable following BA treatment (Fig 6J–6L and S13 Table). These results suggest that BA treatment can ameliorate abnormal developmental competence resulting from abnormal oocyte maturation induced by Bru, an Nrf2 inhibitor.

### Effects of BA on Bru-exposed porcine oocytes under oxidative stress

To assess the antioxidant effect of BA on Bru-treated oocytes, intracellular ROS and GSH levels were measured. Elevated ROS levels induced by Bru, compared to the control, were significantly ameliorated by BA treatment (Fig 7A and 7B). Furthermore, GSH levels decreased by Bru exposure, compared to the control, were significantly restored by BA treatment (Fig 7C and 7D). Notably, the decreased expression of antioxidant genes induced by Bru was significantly increased by BA treatment (Fig 7E). These results suggest that BA can ameliorate increased intracellular ROS levels in Bru-treated oocytes through the regulation of antioxidant gene expression, which is controlled by the Nrf2/Keap1 signaling pathway.

## Discussion

The production of high-quality oocytes is the first step toward successful embryonic development for *in vitro* production or the production of transgenic animals [24]. However, oocyte maturation failure during IVM can substantially hinder embryonic development, implantation, and pregnancy maintenance [25]. Therefore, the enhancement of oocyte quality during IVM constitutes a major challenge for successful *in vitro* production. Because various factors influence oocyte maturation, optimization of the IVM medium environment is the primary obstacle to achieving successful oocyte maturation [26]. Despite extensive effort, the oocyte maturation rate *in vitro* remains lower than the rate achieved *in vivo* [27]. Among various influential factors, high oxidative stress is a major contributor to maturation failure *in vitro* [28]; higher oxidative stress levels are detected during IVM compared with *in vivo*, leading to oocyte maturation failure [7]. Therefore, we utilized the antioxidant BA to experimentally mitigate oxidative stress during IVM; we found that it enhanced meiotic maturation rates and subsequent embryonic development. Furthermore, we explored the mechanisms underlying BA effects on oocytes to obtain key supporting data that will help to improve oocyte maturation rates.

BA is a naturally occurring pentacyclic triterpene that is found in diverse plants, particularly white birch bark and rosemary [29]. BA exhibits anti-inflammatory, antibacterial, and antitumor effects [30], and is notably recognized as an anticancer agent due to its cytotoxic effects at

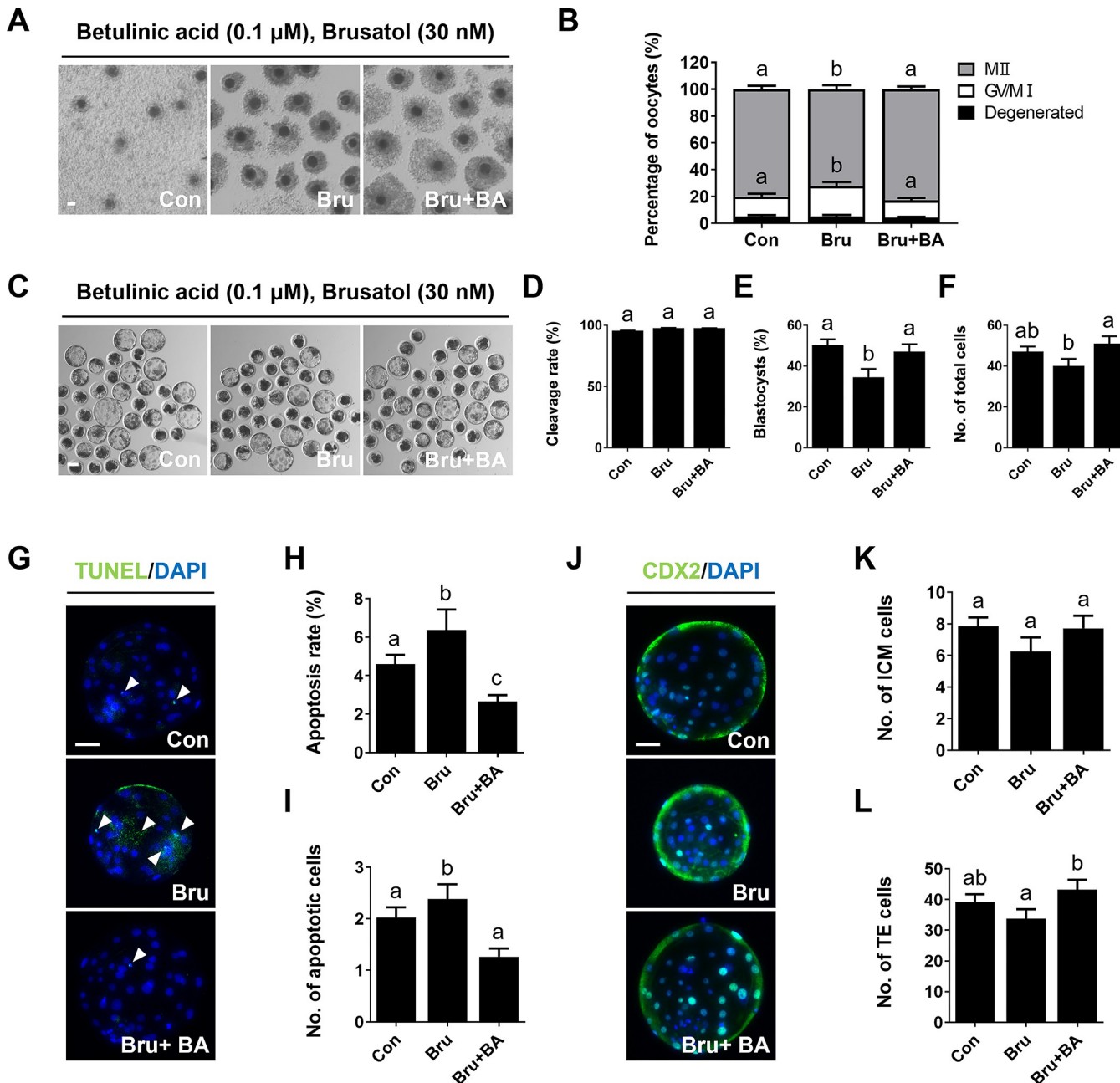

**Fig 6. BA treatment restores oocytes maturation impaired by Bru. A** Bright-field images of oocytes after IVM. Bar = 100 μm. **B** Proportions of different stages of nuclear maturation (0.0 μM BA, n = 391; Bru, n = 395; Bru+BA, n = 382). **C** Bright-field images of BA-treated oocytes at 6 days of culture after parthenogenetic activation. Bar = 100 μm. **D–F** Cleavage rate, blastocyst formation rate, and total cell number measurements (0.0 μM BA, n = 154; Bru, n = 132; Bru+BA, n = 152). **G** Representative images of TUNEL labeling in blastocysts. Merged images show cells positive for DAPI (blue) and TUNEL (green; white arrows). Bar = 50 μm. **H, I** Apoptosis rates and cell numbers (n = 26 per group). **J** Representative images of CDX2 labeling in blastocysts. Merged images show cells positive for DAPI (blue) and CDX2 (green). Bar = 50 μm. **K, L** Numbers of TE and ICM cells (n = 24 per group). Data are means of three independent experiments; different letters indicate significant differences (P < 0.05).

high concentrations [12] and antioxidant effects at low concentrations [31]. The antioxidative effects of BA primarily involve the upregulation of antioxidant gene expression. In yeast, BA mitigates oxidative stress induced by $H_2O_2$ by reducing ROS levels [32], and in mice, BA prevents oxidative damage in the liver and spleen by upregulating the expression of antioxidant

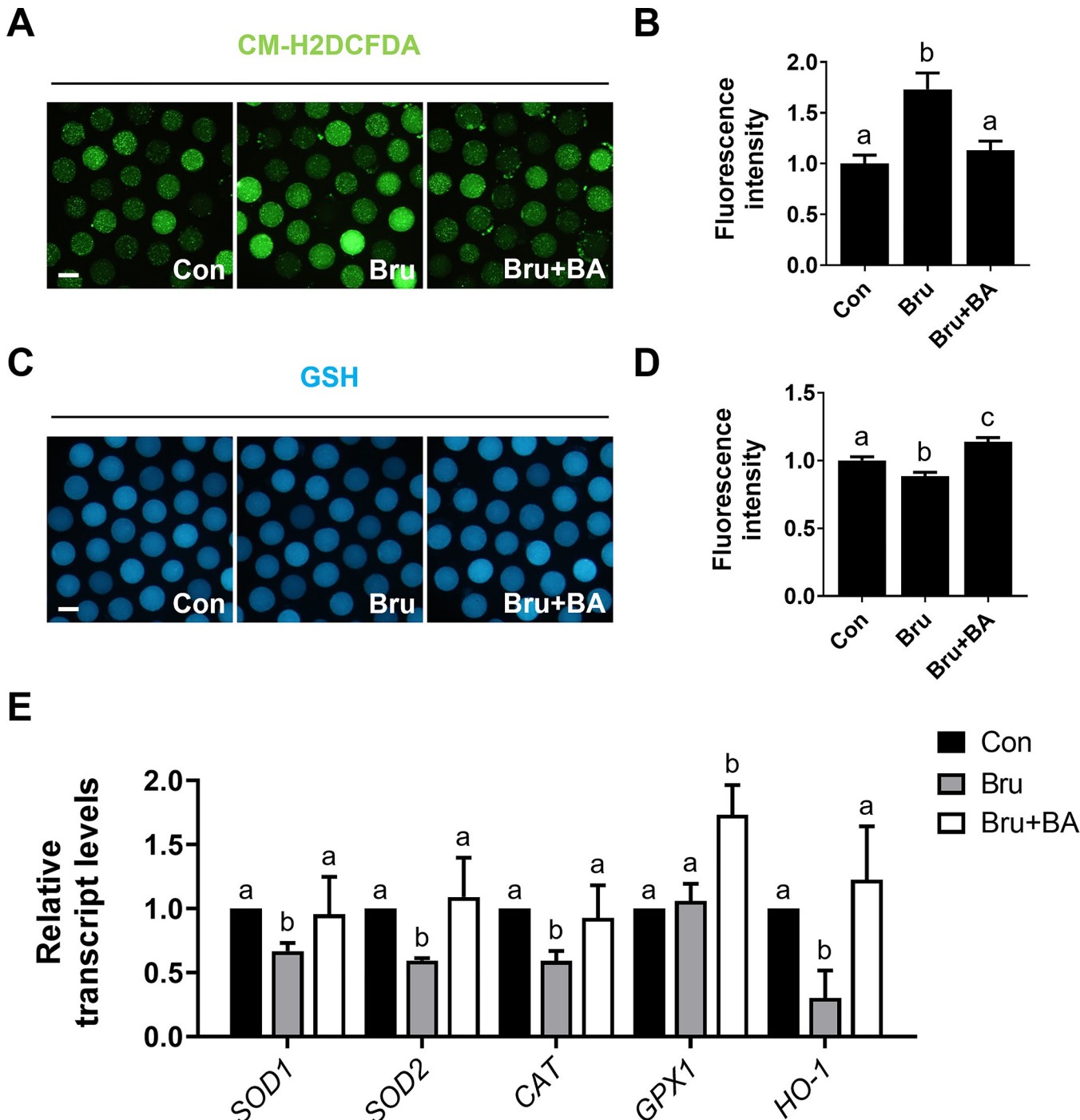

**Fig 7. BA treatment restores impaired oxidative stress caused by Bru. A** Representative images of CM-H2DCFDA staining (green) in oocytes after 44 h of IVM. Bar = 100 μm. **B** Measurements of ROS fluorescence intensity (n = 40 per group). **C** Representative images of GSH staining (blue) in oocytes after 44 h of IVM. Bar = 100 μm. **D** Measurements of the GSH fluorescence intensity (n = 40 per group). **E** Transcription levels derived from qRT-PCR analysis of antioxidant genes in oocytes (n = 3 per group). Data are means of three independent experiments; different letters indicate significant differences (P < 0.05).

genes including *SOD1*, *SOD2*, *CAT*, *GPX1*, and *Nrf2* [14,15]. Therefore, we hypothesized that BA treatment during IVM could improve oocyte quality by reducing oxidative stress. Our findings demonstrated that BA treatment during IVM significantly increased the proportion

of matured oocytes, and BA-treated oocytes exhibited improved early embryonic development parameters such as blastocyst formation rates, cell numbers, and survival rates during preimplantation stages. These results suggest that BA treatment may enhance oocyte quality, improving the success of early embryonic development during preimplantation. Notably, even 1.0 μM BA led to slightly decreased meiotic maturation and blastocyst formation rates compared to 0.1 μM BA. A previous study showed that high BA concentrations cause G2/M arrest in human cancer cells [33]; thus, it is possible that the decreased maturation levels observed following high-concentration BA treatment were attributable to the cytotoxic effects of BA. These findings suggest that 0.1 μM BA is the optimal concentration for porcine oocyte maturation and highlights the potential of BA as an antioxidant agent in porcine IVM systems.

We hypothesize that the improved oocyte maturation levels and embryonic development observed following BA treatment were the result of its antioxidant effects. To confirm this hypothesis, we examined the levels of ROS and GSH, which are oxidative stress markers [8,34], in matured porcine oocytes. The activity of oxygen molecules during aerobic metabolism inevitably leads to ROS formation [35]. Although ROS play essential roles as signaling molecules in cells, their accumulation can induce oxidative stress [36,37]. Conversely, GSH plays a critical role in protecting cells from oxidative stress [38] and is a major factor determining oocyte quality [39]. As shown in Fig 2, BA treatment reduced ROS levels and increased GSH levels in matured porcine oocytes after 44 h of IVM. To elucidate the mechanism of BA antioxidant activity underlying these effects, we exposed BA-treated oocytes to the ROS activator $H_2O_2$ [40,41]. As shown in Fig 3 and S6 Table, the decreased meiotic maturation, cleavage, and blastocyst formation rates induced by $H_2O_2$ were mitigated by BA treatment. Additionally, BA alleviated the increased apoptosis rate induced by $H_2O_2$. Intriguingly, TE and ICM cell numbers were significantly decreased and increased, respectively, by BA treatment compared to the control condition. Furthermore, the increased ROS levels and decreased GSH levels induced by $H_2O_2$ were also mitigated by BA treatment. Surprisingly, BA treatment upregulated antioxidant-related genes that are primarily regulated by Nrf2, a major enzyme controlling the cellular oxidative stress defense system [42]. Therefore, we additionally assessed the transcription levels of *Nrf2* and *Keap1* following $H_2O_2$ and BA treatment (Fig 4). Reduced *Nrf2* transcription levels and increased *Keap1* transcription levels induced by $H_2O_2$ were mitigated by BA treatment. Together, these results suggest that the antioxidant effects of BA are related to the Nrf2/Keap1 signaling pathway.

Nrf2 is a central transcription factor that primarily upregulates antioxidant response elements, inducing the expression of antioxidant enzymes to protect cells from oxidative stress [43]. Nrf2 directly binds to Keap1, a redox-sensitive E3 ubiquitin ligase substrate adaptor [44,45]. Combined with Nrf2, this complex controls the stability and accumulation of Nrf2. The direct binding of Keap1 and Nrf2 prevents ubiquitination of lysine residues in the NRF2 Neh2 domain, leading to the proteasomal degradation of NRF2 [46]. However, under stress conditions, Keap1 allows Nrf2 to escape ubiquitination by cysteine, after which it can accumulate within the cell and translocate to the nucleus to initiate antioxidant gene transcription [46]. Therefore, the inactivation of Keap1 or overexpression of Nrf2 can induce the transcriptional expression of antioxidant genes and enzymes, such as *SOD*, *CAT*, and *HO-1*, protecting cells from oxidative stress [47]. In our study, these genes were regulated by BA treatment (Figs 2 and 4). Therefore, we used the Nrf2 inhibitor Bru to confirm the effects of BA treatment on the Nrf2/Keap1 signaling pathway [18,48]. Bru is a widely used Nrf2 inhibitor that has been extensively studied as an Nrf2 inhibitor; it has been utilized as an anticancer agent due to its inhibition of Nrf2 [48,49]. Bru specifically suppresses Nrf2 protein expression by blocking the translation of Nrf2 mRNA [50], which causes severe critical damage to oocytes via Nrf2 signaling disruption. These effects collectively block the expression of antioxidant genes that protect

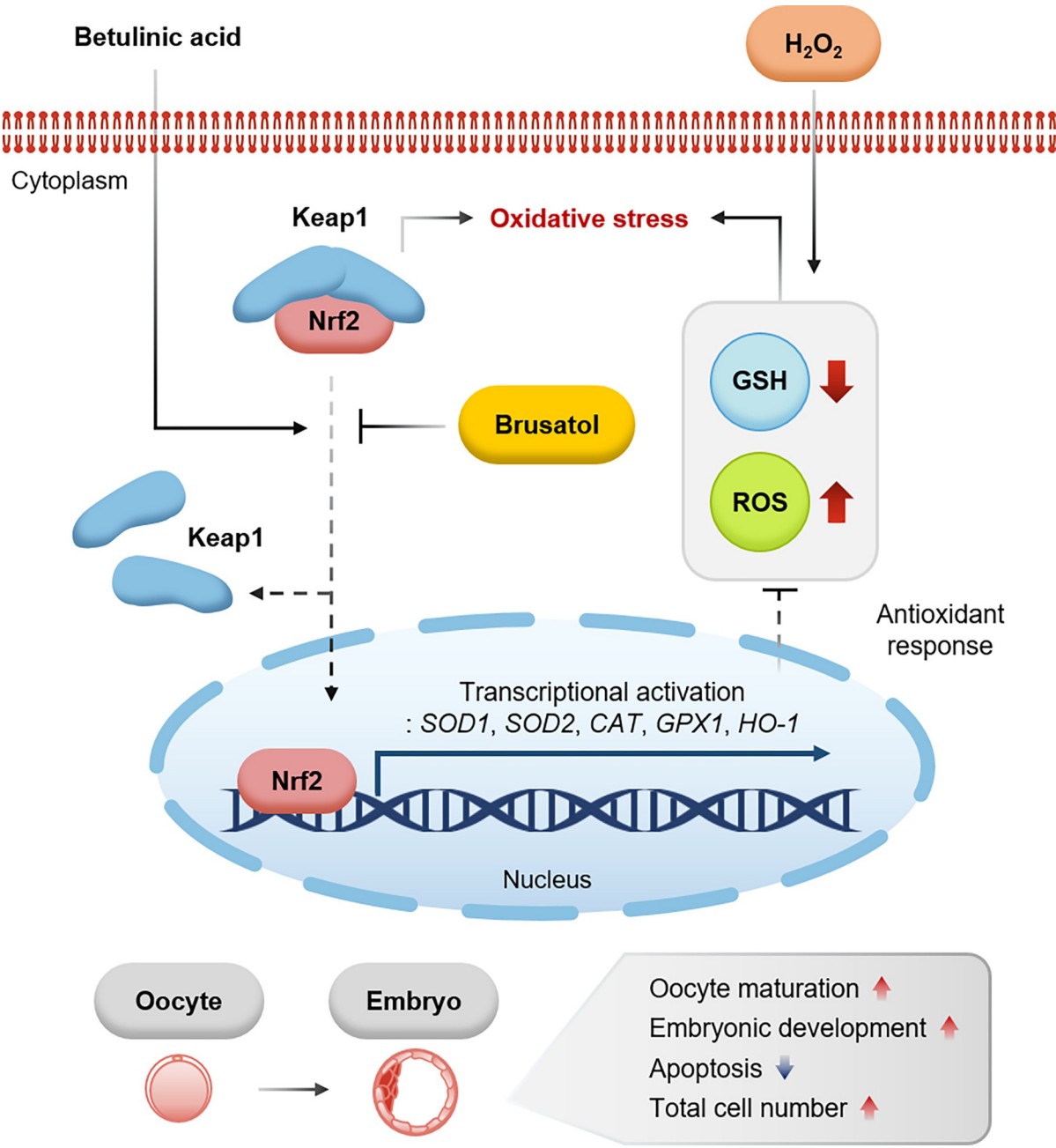

**Fig 8. Graphical overview of the effect of BA treatment on porcine oocyte maturation.** BA treatment increased the proportion of MII oocytes, enhanced developmental rates, cell numbers, trophectoderm rates, and cell survival compared to control. Additionally, BA-treated oocytes exhibited reduced levels of ROS and elevated levels of glutathione, accompanied by enhanced expression of antioxidant genes. Especially, BA treatment mitigated the negative effects of $H_2O_2$-induced ROS activation and the Nrf2 inhibitor, brusatol, on meiotic maturation and oocyte quality. These results suggest that BA affects beneficial effects on the maturation of porcine oocytes, which can be attributable to the activation of the Nrf2/Keap1 signaling pathway by BA.

cells from ROS. The application of Bru at concentrations above 50 nM leads to oocyte death [51,52]. As shown in Fig 5, Nrf2 levels were significantly increased by BA treatment during IVM, accompanied by decreased Keap1 levels, and the decrease in Nrf2 levels induced by Bru was significantly mitigated by BA treatment. Furthermore, as observed with $H_2O_2$, decreased

meiotic maturation, development competence, and total cell numbers induced by Bru exposure were significantly mitigated by BA treatment. Additionally, decreased survival rates and TE cell numbers in blastocysts of the Bru-treated group were mitigated, indicating improved oocyte quality (Fig 6 and S10 Table). These results suggest that BA can alleviate the negative effects of Bru. Finally, we evaluated the effect of BA on Bru-treated oocytes. Increased ROS levels and decreased GSH levels induced by Bru were significantly mitigated by BA treatment, and the transcription levels of antioxidant genes regulated by Bru were significantly mitigated or even increased by BA treatment compared to the control. These results demonstrate that the antioxidant effects of BA are attributable to Nrf2/Keap1 signaling pathway regulation. However, despite the positive effects of BA on oocyte maturation *in vitro*, it remains unclear whether the consistent intake of BA has beneficial effects on oocytes *in vivo*. Therefore, further research is needed regarding the long-term effects of BA on embryonic development, which is a crucial consideration when evaluating potential clinical applications for BA.

## Conclusions

In conclusion, this study is the first to report antioxidant effects of BA on meiotic maturation and developmental competence in porcine oocytes, which were attributable to the activation of the Nrf2/Keap1 signaling pathway by BA (Fig 8). These findings will contribute to the development of methods for improving IVM systems for oocyte maturation, and the interaction between antioxidants and the Nrf2/Keap1 signaling pathway reported in this study may provide insights into their roles in porcine oocyte maturation.

## Supporting information

**S1 Table. Primer sequences used for qRT-PCR.**
(DOCX)

**S2 Table. Effect of BA treatment on nuclear maturation of porcine oocytes.**
(DOCX)

**S3 Table. Developmental competence of BA treated porcine oocytes.**
(DOCX)

**S4 Table. Effects of BA treatment during IVM on cell survival in blastocyst.**
(DOCX)

**S5 Table. Effects of BA treatment during IVM on number of TE and ICM cells in blastocyst.**
(DOCX)

**S6 Table. Effect of BA treatment on nuclear maturation of $H_2O_2$-exposed porcine oocytes.**
(DOCX)

**S7 Table. Developmental competence of BA treatment on $H_2O_2$-exposed porcine oocytes.**
(DOCX)

**S8 Table. Effects of BA on $H_2O_2$-exposed oocytes for cell survival in blastocyst.**
(DOCX)

**S9 Table. Effects of BA on $H_2O_2$-exposed oocytes for number of TE and ICM cells in blastocyst.**
(DOCX)

**S10 Table. Effect of BA treatment on nuclear maturation of Bru-exposed porcine oocytes.**
(DOCX)

**S11 Table. Developmental competence of BA treatment on Bru-exposed porcine oocytes.**
(DOCX)

**S12 Table. Effects of BA on Bru-exposed oocytes for cell survival in blastocyst.**
(DOCX)

**S13 Table. Effects of BA on Bru-exposed oocytes for number of TE and ICM cells in blastocyst.**
(DOCX)

## Author Contributions

**Conceptualization:** Sun-Uk Kim, Bong-Seok Song.

**Data curation:** Min Ju Kim, Hyo-Gu Kang, Se-Been Jeon, Pil-Soo Jeong, Bong-Seok Song.

**Formal analysis:** Min Ju Kim, Hyo-Gu Kang.

**Funding acquisition:** Sun-Uk Kim.

**Investigation:** Min Ju Kim, Hyo-Gu Kang, Ji Hyeon Yun.

**Methodology:** Se-Been Jeon.

**Project administration:** Bo-Woong Sim, Sun-Uk Kim, Seong-Keun Cho, Bong-Seok Song.

**Resources:** Se-Been Jeon, Ji Hyeon Yun, Pil-Soo Jeong, Bo-Woong Sim.

**Software:** Ji Hyeon Yun.

**Supervision:** Seong-Keun Cho, Bong-Seok Song.

**Validation:** Min Ju Kim, Hyo-Gu Kang.

**Writing – original draft:** Min Ju Kim, Hyo-Gu Kang.

**Writing – review & editing:** Seong-Keun Cho, Bong-Seok Song.

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
