## [Decision Letter · Decision Letter 0]

6 Aug 2024

PONE-D-24-25068The antioxidant betulinic acid enhances porcine oocyte maturation through Nrf2/Keap1 signaling pathway modulationPLOS ONE

Dear Dr. Song,

Thank you for submitting your manuscript to PLOS ONE. After careful consideration, we feel that it has merit but does not fully meet PLOS ONE’s publication criteria as it currently stands. Therefore, we invite you to submit a revised version of the manuscript that addresses the points raised during the review process.

The results need to be counted and analyzed correctly, and another Nrf2 inhibitor is needed to confirm the involvement of the Nrf2/Keap1 pathway.

We look forward to receiving your revised manuscript.

Kind regards,

Meijia Zhang

Academic Editor

PLOS ONE

2. In your Methods section, please report the source of Betulinic acid used for your study.

Reviewers' comments:

Reviewer's Responses to Questions

**Comments to the Author**

1. Is the manuscript technically sound, and do the data support the conclusions?

Reviewer #1: Yes

Reviewer #2: Yes

2. Has the statistical analysis been performed appropriately and rigorously? 

Reviewer #1: Yes

Reviewer #2: Yes

3. Have the authors made all data underlying the findings in their manuscript fully available?

Reviewer #1: Yes

Reviewer #2: Yes

4. Is the manuscript presented in an intelligible fashion and written in standard English?

Reviewer #1: No

Reviewer #2: Yes

5. Review Comments to the Author

Reviewer #1: Excess levels of reactive oxygen species (ROS) are a major cause of developmental defects in embryos in vitro. Here, authors found that BA plays an effective role as an antioxidant in porcine oocyte maturation through adjusting the Nrf2/Keap1 signaling pathway. Their finding wound benefit for oocyte maturation and embryonic development in vitro.

Main comment:

(1) Authors claimed that BA plays its role by the Nrf2/Keap1 signaling pathway. However, only a potent Nrf2 inhibitor is not enough. Also, BA still recovered the inhibitory effects of Bru in author’s experiment. What dose of Bru can completely suppress the effects of BA?

(2) There are some flaws in the figure, such as no scale bar in some images; The columns of some figures are distorted.

Reviewer #2: The study investigates the effect of betulinic acid (BA), a natural antioxidant found in white birch bark, on the maturation of porcine oocytes in vitro. The research demonstrates that BA significantly enhances oocyte maturation rates, improves embryonic development, increases cell survival, and reduces oxidative stress by modulating the Nrf2/Keap1 signaling pathway. These findings provide valuable insights into optimizing in vitro maturation systems for oocytes. Overall, this study offers promising evidence for the potential application of BA in improving oocyte quality and reproductive outcomes in assisted reproductive technologies, which could be of great interest to the readers of this journal given its implications for advancing reproductive biology and biomedicine.This article should be revised according to the following comments before publication:

1. In Figure 1 D and K, the columns should be labeled with letters even though they show no statistical difference. Similar issues appear in other figures. Please check it.

2. In Figure 3G, the images of blastocysts from different treatments show similar TUNEL staining patterns. The authors should consider using more representative images.

3. In Figure 3J, K, and L, why does the H2O2 treatment seem to increase the inner cell mass cell number according to Figure 3J? Does H2O2 promote ICM cell proliferation?

4. Line 290 mentions brusatol as an Nrf2 inhibitor to confirm the involvement of the Nrf2/Keap1 pathway. Could you provide more details on the specificity of brusatol and any potential off-target effects that might influence the interpretation of your results?

5. The discussion section of this paper does not address the long-term effects of BA on embryonic development. The authors should consider this issue when applying BA for clinical applications.

6. PLOS authors have the option to publish the peer review history of their article (what does this mean?). If published, this will include your full peer review and any attached files.

Reviewer #1: No

Reviewer #2: No

---

## [Author Response · Author response to Decision Letter 0]

30 Aug 2024

We have provided thorough responses to the questions you requested to be revised, and we kindly ask for your approval for the publication of our manuscript.

---

## [Decision Letter · Decision Letter 1]

17 Sep 2024

PONE-D-24-25068R1The antioxidant betulinic acid enhances porcine oocyte maturation through Nrf2/Keap1 signaling pathway modulationPLOS ONE

Dear Dr. Song,

Thank you for submitting your manuscript to PLOS ONE. After careful consideration, we feel that it has merit but does not fully meet PLOS ONE’s publication criteria as it currently stands. Therefore, we invite you to submit a revised version of the manuscript that addresses the points raised during the review process.

We look forward to receiving your revised manuscript.

Kind regards,

Meijia Zhang

Academic Editor

PLOS ONE

Journal Requirements:

**Additional Editor Comments:**

An error in the title of X axis of Fig 1B.

Reviewers' comments:

Reviewer's Responses to Questions

**Comments to the Author**

1. If the authors have adequately addressed your comments raised in a previous round of review and you feel that this manuscript is now acceptable for publication, you may indicate that here to bypass the “Comments to the Author” section, enter your conflict of interest statement in the “Confidential to Editor” section, and submit your "Accept" recommendation.

Reviewer #1: All comments have been addressed

Reviewer #2: All comments have been addressed

2. Is the manuscript technically sound, and do the data support the conclusions?

Reviewer #1: Yes

Reviewer #2: Yes

3. Has the statistical analysis been performed appropriately and rigorously? 

Reviewer #1: Yes

Reviewer #2: Yes

4. Have the authors made all data underlying the findings in their manuscript fully available?

Reviewer #1: Yes

Reviewer #2: Yes

5. Is the manuscript presented in an intelligible fashion and written in standard English?

Reviewer #1: Yes

Reviewer #2: Yes

6. Review Comments to the Author

Reviewer #1: (No Response)

Reviewer #2: All comments have been addressed. However, an error was found in the title of x axies of Figure 1B. Please revise it.

7. PLOS authors have the option to publish the peer review history of their article (what does this mean?). If published, this will include your full peer review and any attached files.

Reviewer #1: No

Reviewer #2: No

---

## [Author Response · Author response to Decision Letter 1]

19 Sep 2024

We greatly appreciate the reviewers' agreement with many aspects of our research findings. We have made the requested revisions to the figure 1 and uploaded the updated files. Kindly review them and proceed with the approval for publication

---

## [Decision Letter · Decision Letter 2]

26 Sep 2024

The antioxidant betulinic acid enhances porcine oocyte maturation through Nrf2/Keap1 signaling pathway modulation

PONE-D-24-25068R2

Dear Dr. Song,

We’re pleased to inform you that your manuscript has been judged scientifically suitable for publication and will be formally accepted for publication once it meets all outstanding technical requirements.

Kind regards,

Meijia Zhang

Academic Editor

PLOS ONE

Additional Editor Comments (optional):

Reviewers' comments:

Reviewer's Responses to Questions

**Comments to the Author**

1. If the authors have adequately addressed your comments raised in a previous round of review and you feel that this manuscript is now acceptable for publication, you may indicate that here to bypass the “Comments to the Author” section, enter your conflict of interest statement in the “Confidential to Editor” section, and submit your "Accept" recommendation.

Reviewer #2: All comments have been addressed

2. Is the manuscript technically sound, and do the data support the conclusions?

Reviewer #2: Yes

3. Has the statistical analysis been performed appropriately and rigorously? 

Reviewer #2: Yes

4. Have the authors made all data underlying the findings in their manuscript fully available?

Reviewer #2: Yes

5. Is the manuscript presented in an intelligible fashion and written in standard English?

Reviewer #2: Yes

6. Review Comments to the Author

Reviewer #2: All comments have been addressed. There are no more comments for the authors. This paper should be considered accepted.

7. PLOS authors have the option to publish the peer review history of their article (what does this mean?). If published, this will include your full peer review and any attached files.

Reviewer #2: No

---

## [Editor Report · Acceptance letter]

2 Oct 2024

PONE-D-24-25068R2 

PLOS ONE

Dear Dr. Song, 

I'm pleased to inform you that your manuscript has been deemed suitable for publication in PLOS ONE. Congratulations! Your manuscript is now being handed over to our production team.

Kind regards, 

on behalf of

Dr. Meijia Zhang 

Academic Editor

PLOS ONE